# Effect of Multiple Sclerosis and Its Treatments on Male Fertility: Cues for Future Research

**DOI:** 10.3390/jcm10225401

**Published:** 2021-11-19

**Authors:** Claudia Massarotti, Elvira Sbragia, Irene Gazzo, Sara Stigliani, Matilde Inglese, Paola Anserini

**Affiliations:** 1Department of Neurosciences, Rehabilitation, Ophthalmology, Genetics, Maternal and Child Health, University of Genoa, 16128 Genova, Italy; claudia.massarotti@unige.it (C.M.); irigazzo@libero.it (I.G.); 2Academic Unit of Obstetrics and Gynecology, IRCCS Ospedale Policlinico San Martino, 16132 Genova, Italy; 3Center of Excellence for Biomedical Research and Department of Neurosciences, Rehabilitation, Ophthalmology, Genetics, Maternal and Child Health, University of Genoa, 16128 Genova, Italy; elvis.sbragia@gmail.com (E.S.); m.inglese@unige.it (M.I.); 4Physiopathology of Human Reproduction Unit, IRCCS Ospedale Policlinico San Martino, 16132 Genova, Italy; sara.stigliani@hsanmartino.it; 5IRCCS Ospedale Policlinico San Martino, 16132 Genova, Italy

**Keywords:** multiple sclerosis, male fertility, sperm analysis, gonadotoxicity, reproduction and chronic diseases, fertility and multiple sclerosis

## Abstract

Multiple sclerosis is a chronic disease that may lead to different types of symptoms and disabilities. with the better quality of life and decreased disability due to early diagnosis and the availability of disease-modifying therapies (DMTs), the treating physician is increasingly asked to counsel patients on its effects on fertility and reproduction. In particular, reproductive issues are still scarcely studied and discussed in men. Among the still open questions are the following: (a) Does multiple sclerosis cause infertility per sè? (b) Is multiple sclerosis correlated with conditions that increase the risk of infertility? (c) Do DMTs or other therapies for multiple sclerosis impact gonadal function in men? The aim of this review is to provide an overview on the available literature data about the reproductive issues unique to men with multiple sclerosis, underlining the numerous areas where evidence is lacking and, therefore, the priorities for future research.

## 1. Introduction

Multiple sclerosis (MS) is a chronic inflammatory, demyelinating, and neurodegenerative disease of the central nervous system (CNS) [1,2], affecting 2.8 million people in 2020 [3]. Recent data [2] estimate the 2020 global prevalence to be 35.9 (95% CI: 35.87, 35.95) per 100,000 people. It is the most common cause of non-traumatic disability in young adults [4,5].

Its etiology is still unknown, as with most autoimmune diseases. Nevertheless, in recent decades, it has been proposed that several environmental factors (e.g., ultraviolet B light exposure, vitamin D levels, Epstein–Barr virus infection, obesity, and smoking) [6,7] may trigger MS within predisposed subjects. In particular, some genetic polymorphisms, mostly involving immune pathway genes [2], have been described as being associated with a higher risk of developing MS. The most significant genetic risk factor is the HLA DRB1*1501 haplotype [2,6,7]. Moreover, sex hormones possibly interfere with disease onset [8,9].

The typical pathological hallmark of MS is represented by distinct CNS lesions, easily recognized in white matter through magnetic resonance imaging (MRI), leading to plaques whose genesis involves the mechanism of demyelination, inflammation, and glial reaction [2,6,10]. The inflammatory infiltrates contain T-lymphocytes, dominated by MHC class I restricted CD8+ T-cells; B-cells and plasma cells are also present, although in much lower numbers [10]. Lesions of MS are typically located in periventricular, cortical, juxtacortical, and infratentorial brain regions and in the spinal cord [11]. A new lesion, in particular in an eloquent area, leads to corresponding symptoms that vary depending on location and size of the lesions. Moreover, macroscopic (or MRI-visible) lesions are just the tip of the iceberg; many more lesions can be seen at the microscopic level, and even more in deep and cortical grey matter, underlining the so-called “clinical-radiological paradox”, where clinical disability evidence does not always present a clear association with structural imaging studies.

MS is clinically characterized in most cases by the subacute onset of focal or multifocal neurological symptoms due to the involvement of different CNS systems (optic nerves, both hemispheres, brainstem, cerebellum, and spinal cord), which are either fully or partially reversible [6]. These episodes develop over hours to days and then reach a plateau lasting several weeks, followed by a gradual recovery that occurs spontaneously or with a high dose of steroidal therapy. These focal episodes are defined as “relapses”. When MS presents with multiple new focal symptoms occurring sequentially and ceasing afterwards, the onset of relapsing–remitting MS is signaled. Each relapse corresponds to at least one symptomatic lesion, but generally within a clinical attack, approximately 10 “asymptomatic” lesions are noted in MRI [6]. Later on in the course of the disease (usually 10–20 years from onset), after a period of tumultuous, inflammatory disease, most patients develop a slow worsening of their neurological condition with no distinct transition in between, shaping the so-called “progressive course”. In some patients (5–15%), MS starts as progressive from onset [2,6].

MS, as with most autoimmune diseases, is more common in women than in men, with the female-to-male ratio varying from 2:1 to 3:1 [12]. The age of onset increases after adolescence, reaching a peak at the third decade [13] in relapsing–remitting forms, between 25 and 35 years [7], and at the fifth decade [13] in primary–progressive forms. Nevertheless, some gender issues do exist. In particular, men receive diagnosis of MS at an older age than women [8,9,13], and the course of the disease is prognostically worse in men, with a higher tendency towards motor and disabling relapses and towards a progressive course [8,9,13].

However, MS incidence remains higher in both men and women of reproductive age. Therefore, especially as a consequence of earlier diagnoses and more efficacious therapies, it is common to treat patients with reproductive desires. In women, we know that the disease improves during pregnancy and worsens after birth [14,15], leading scientists to hypothesize a connection with sex hormones levels [9]. However, the possible effects of the disease and of disease-modifying therapies (DMTs) on fertility and pregnancy are not yet comprehensively determined in the literature [16], and in particular, they are severely under-investigated in men. Indeed, there are still unanswered questions on male fertility and MS; among them are the following: (a) Does MS cause infertility per se? (b) Is MS correlated with conditions that increase the risk of infertility? (c) Do DMTs or other therapies for MS impact gonadal function in men?

## 2. Methods

The main objective of this narrative review is to provide an overview of the possible reproductive issues unique to men with multiple sclerosis, with the aim of providing clinicians with a useful tool for counseling and highlighting possible research priorities. A review of literature published in English until July 2021 was conducted on PubMed, Scopus, and Google Scholar, using the search terms “multiple sclerosis” and/or “disease modifying agents” and “gonadotoxicity” and/or “fertility” and/or “male fertility”. In addition to that, reference lists of primary and review articles were reviewed for additional publications. Selected articles were verified by two separate authors and discrepancies were discussed within the team.

## 3. Results

### 3.1. Reproductive History of Men with Multiple Sclerosis

The issue of fertility in men with MS has not been exhaustively studied. There are currently no studies in the literature evaluating the prospective time to conceive of men with MS; only indirect information from population registries is available. Epidemiology studies have reported a decreased number of pregnancies in partners of males with MS. In particular, a Swedish population study described how men with a new diagnosis of MS were less likely to have had children in the 5 years before diagnosis [17]. A similar study in the Danish population reached similar conclusions some years before: out of more than 4 million Danish men, those who had become parents in the 5 years before had a significantly lower risk of being diagnosed with MS [18]. Of course, these results do not prove a correlation between the disease and reduced fertility, since these data are not calculated on the total number of men who searched for a pregnancy: those who had not had children may have not tried to conceive to begin with. Additional indirect information may be inferred from examining infertile couples. In 2008, a study reported worse seminal parameters (total count, motility, and morphology) in 68 MS patients, compared to 48 age-matched controls [19]. Data from the Danish population registry showed how the prevalence of multiple sclerosis was higher (odds ratio (OR) = 1.61, 95% confidence interval (CI): 1.04–2.51) in male partners of couples treated for male infertility than in male partners of couples whose infertility was not linked to a male factor [20].

### 3.2. Possible Rationale for Reduced Fertility in Men with Multiple Sclerosis

It is known that MS can directly affect sexual function in men, depending on the location of the plaques. Erectile function is controlled by parasympathetic fibers originating from S2 to S4; ejaculation depends on the coordinated action of thoracolumbar sympathetic fibers from segments T10–L2 and somatic fibers from S2 to S4. Central regulation also plays a role in ejaculation [21]. Impotence in MS can be ascribed to a suprasacral, parasympathetic, or peripheral autonomic lesion. Up to 70% of men with MS present with erection dysfunction, and up to 50% of men present with alterations in ejaculation [22,23]. Moreover, some medications like benzodiazepine or tricyclic antidepressants, used as symptomatic therapies, could affect sexual function as well. Possible therapeutic solutions include phosphodiesterase inhibitors (i.e., Sildenafil) for erectile dysfunction, assisted ejaculation, or surgical sperm retrieval coupled with assisted reproduction techniques [24,25,26]. Multiple sclerosis could also disrupt hypothalamus–pituitary function, with reduced levels of sex hormones as a consequence of central neurological damage [19,27,28,29,30]. Indeed, hypogonadotropic hypogonadism has been described, especially among male with more progressive disease [19].

Pathogenetic causes of MS may also have a cross sectional or longitudinal link with infertility. MS is defined as an autoimmune response to unidentified antigen(s) in genetically susceptible individuals [31]. Genetic predisposition, autoimmunity, and chronic inflammation could all be linked to male infertility pathogenesis. For example, genetic susceptibility to infertility has been hypothesized due to increasing evidence of a link between it and risk of degenerative diseases, but current data do not support a correlation with MS specifically [32]. A correlation between male infertility and autoimmunity has also been reported: various autoimmune conditions, such as type I diabetes mellitus or Crohn’s disease, were found to be more prevalent in infertile men [33,34]. Chronic inflammation has been linked to both MS and male infertility [20]. In particular, chronic inflammation was reported in men with low levels of testosterone [35], a connection worth examining when discussing MS. Bove et al. described a strong positive association between testicular hypofunction and subsequent MS (rate ratio = 4.62, 95% CI = 2.3–8.24, *p* < 0.0001), and low testosterone levels have been associated with higher disability in patients already diagnosed with MS [36]. These findings are in agreement with the fact that age at diagnosis in men is higher than in women, and that this age corresponds with the age when testosterone levels start to decline [37]. However, it is not possible to discriminate whether there is a causal relationship, a longitudinal association, or if the results are only a casualty [38]. An increased risk of MS has been reported in transgender women after hormonal transition that suppresses testosterone levels [39], suggesting a protective role of the hormone.

Oxidative stress, linked to chronic inflammation, plays a role in MS [40] and may be another cause of worse sperm quality, since it is well known that human semen is susceptible to an imbalance between reactive oxygen species (ROS) and antioxidants. In particular, spermatozoa’s plasma membranes are rich in polyunsaturated fatty acids and have low concentrations of scavenging enzymes, and therefore, their ROS-induced peroxidation causes a decrease in mobility. Moreover, ROS may cause direct damage to mitochondria, damaging sperm DNA and suppressing sperm capacitation and acrosome reaction [41,42].

In summary, men with MS could be subfertile due to a direct negative impact of the disease on erection and/or ejaculation or due to concurrent hypogonadism, both of central (direct action on the hypothalamus–pituitary–testis axis) and peripheral (chronic inflammation and autoimmunity) origin. To these possible causes, we need to add the gonadotoxic risk of the received medications.

Table 1 and Figure 1 summarize the possible causes of infertility in male patients with MS.

### 3.3. Gonadotoxicity of Disease-Modifying Therapies in Male Patients

Data about the effect of DMTs on male reproduction, both as impact on spermatogenesis and as risk connected with fathering a child during therapies, are sparse and inconclusive [16]. In particular, no other data except those from preclinical animal studies reporting on prescribing information are available for most of the newest drugs approved for treatment [43]. The fact that patients are frequently subjected to the newest second-line therapies after the failure of first-line ones further complicates our ability to quantify the absolute damage of specific treatments.

Table 2 reports what is known about the effect of DMTs on male fertility.

Glatiramer Acetate and Interferon are both immunomodulatory drugs that have been used in the treatment of MS for more than 20 years. Two multicenter prospective studies reported the outcome of newborns fathered by men exposed to these drugs at the time of conception; no increased risk of spontaneous abortion, adverse fetal outcomes, or congenital malformations were reported [44,45]. As for the effects on spermatogenesis, no effect was reported in in vitro and animal registration studies [46], and while the aforementioned studies do not investigate sperm parameters, the reporting of healthy pregnancies and no increase in adverse fetal outcomes are reassuring for fertility as well.

Mitoxantrone (no longer used in clinical practice for the treatment of young men with MS), is a type II topoisomerase inhibitor that disrupts DNA synthesis and DNA repair; it is known to cause oligo or azoospermia in men (usually reversible). Its use therefore carries a strong recommendation of semen cryopreservation before starting the therapy [47,48]. Of note, the evidence is mostly from data of onco-hematological patients using chemotherapy containing Mitoxantrone among other drugs. Three other drugs, used less frequently and off-label, are reported to have an effect on male reproduction: Cyclophosphamide, Azathioprine, and Methotrexate. All three have an FDA warning for fertility and pregnancy on their prescribing information documents. Cyclophosphamide, an alkylating agent that disrupts cell division in rapidly dividing cells, produces permanent damage to the seminiferous epithelium, causing dose-dependent azoospermia (not always reversible) as well as an increased risk of abortion in pregnancies fathered by men treated with this therapy [49,50,51]. Semen cryopreservation before therapy and a washout of 6 months (if fresh sperm is used to conceive) are recommended [52]. Azathioprine, an immunosuppressive agent that acts through its effects as an antagonist of purine metabolism, causes oligospermia in animal studies [53,54], but no alteration of sperm parameters during chronic treatment was reported in small cohorts of men with inflammatory bowel disease [55,56] and rheumatological diseases [57,58]. A washout of 6 months before trying to conceive is advised on prescribing documents. However, the concerns are mainly based on a prescription database study reporting 4 malformations out of 52 children (7.4%) fathered by treated men [59]. The study has several limitations (numerosity, consumption deduced based on filled prescriptions and not on directly collected data), and the same authors found no detrimental effect in a more recent bigger cohort [60]. Methotrexate, a dihydrofolate reductase inhibitor, was reported to cause dose-dependent reversible oligospermia, while other studies described minimal or no effect on spermatogenesis [61]. As for pregnancies, while it is abortifacient in women, no increased risk of complications was reported in pregnancies of babies fathered by men during therapy [62,63,64,65]. However, a 3-month washout period before trying to conceive is recommended.

All the other DMTs were approved for use in MS relatively recently, and very few data about their effect on male reproduction are available.

For example, we do not have enough data to exclude an effect on spermatogenesis and/or pregnancy of the more recently approved monoclonal antibodies, such as Natalizumab, Alemtuzumab, Ocrelizumab, Daclizumab, and Rituximab (off-label). Recently, a case-control study did not find any difference in terms of sperm parameters and gonadal steroid levels between 16 men treated with Natalizumab or Ocrelizumab for 12 months and healthy controls [66]. Since Alemtuzumab targets CD52s, which are also present in the male reproductive tract [67], a negative effect on sperm parameters was hypothesized [68], but a small human study (13 patients) did not find any significant variations after 1, 3, and 6 months of therapy [69].

Teriflunomide is a dihydro-orotate dehydrogenase inhibitor, a key mitochondrial enzyme in the de novo pyrimidine synthesis pathway, which leads to a reduction in proliferation of activated T- and B-lymphocytes without causing cell death. It has an FDA warning for pregnancy in women, since it has been associated with an increased risk of fetal malformation in some reports [70], but others concluded that outcomes were consistent with those of the general population [71,72]. We only have limited data for children fathered by men treated with this drug. Teriflunomide can be detected in semen, but two small studies (22 and 18 pregnancies) did not report an increased risk of malformation or other adverse events in pregnancies from fathers using Teriflunomide [71,73].

Teratogenic risk was described for mothers taking Fingolimod at conception [74], but no data are available for men.

We also have no evidence for the effect of Cladribine (approved for MS in 2019) on male reproduction, but, since it interferes with DNA synthesis, an effect on spermatogenesis has been hypothesized [75]. A washout period of 6 months before conception is suggested in the prescribing information documents.

### 3.4. Gonadotoxicity of Symptomatic and Supporting Therapies

Men with MS often find symptomatic (defined as therapies that aim to improve symptoms without modifying the course of the disease) and supporting therapies (treatment aimed to prevent, control, or relieve complications and/or side effects of the main treatment, to improve the patient’s comfort and quality of life) to be beneficial. However, some of them may have a detrimental effect on fertility (usually reversible). Of note, a relevant limitation to consider is that none of the human studies on the effects of such therapies on male fertility is focused specifically on men affected by MS. However, data extrapolated from published cohorts of men who underwent the same treatments in similar dosages are a good starting point to choose the treatment with the minimum possible effect and to help disentangle the complex interactions among the possible causes of subfertility in men with MS.

Antidepressant medications, which are widely used to treat both mood disorders and neuropathic pain, have been associated with sexual dysfunction (impaired libido, erectile dysfunction) and alteration of semen parameters [76]. Benzodiazepines, selective serotonin reuptake inhibitors (SSRIs), serotonin–norepinephrine reuptake inhibitors (SNRIs), and tricyclic antidepressants (TCAs) have been reported to cause hyperprolactinemia, which may cause hypogonadotropic hypogonadism and lower testosterone levels [77]. Animal and human studies have shown a worsening in sperm parameters using SSRIs [78]. In particular, some human studies have underlined an association between sperm DNA fragmentation and SSRI use: in a randomized, single-blind trial examining 60 men treated with Sertraline or behavioral therapy for 3 months, DNA fragmentation significantly increased in the treatment group, while sperm concentration and normal morphology decreased [79]. Increased DNA fragmentation was also reported in the absence of sperm parameter alterations after short periods of treatment with Paroxetine [80]. The described effects were correlated with the duration of antidepressant use [81].

On the other hand, Bupropion, a norepinephrine–dopamine reuptake inhibitor (NDRI) used to treat depression, showed no effect on sperm parameters in rats [82] and lower sexual side effects in humans compared to SSRIs [83]; thus, it is probably a better fit in depressed or anxious young male patients planning to have children [79]. A recent RCT aimed to investigate differences in DNA fragmentation among 68 healthy males after 6 weeks of treatment with 60 mg Duloxetine or placebo; no significant difference was found between the two groups. Furthermore, semen parameters and serum hormones were not altered by the treatment, showing how Duloxetine, and potentially other SNRIs, may be a good choice for young men desiring to conceive [84].

Scant data exist for the effect of TCAs on spermatogenesis. An animal study examining semen parameters of 40 mice after treatment with venlafaxine, venlafaxine plus ascorbic acid, amitriptyline, amitriptyline plus ascorbic acid, or placebo found decreased parameters and increased oxidative stress in the amitriptyline group [85]. Also, two animal studies suggested a mutagenic effect of amitriptyline on germ cells [86,87].

Antiepileptic drugs, used for the treatment of neuropathic pain in men with MS, may affect male reproduction [88]. In particular, Carbamazepine was associated with altered sexual hormone levels in men with epilepsy, especially decreasing bioactive testosterone [89]. Sperm parameters were reported to be altered as well: Azadi-Pooya et al. found a significant reduction in concentration, motility, and morphology in 8 men after 3 months of Carbamazepine therapy for epilepsy [90]. Reis et al. compared 63 men with epilepsy taking Carbamazepine with 55 healthy controls and found a 10-times higher rate of erectile dysfunction in the first group, even after having controlled for confounders [91].

The effects of delta-9-tetrahydrocannabinol (THC) on human semen are yet to be better quantified. We know that deltha-9-THC can cross the blood–testis barrier in certain individuals [92], but there is no definitive evidence of a deleterious effect on seminal parameters. Some studies have reported lower sperm concentrations and lower total sperm counts in marijuana users: a Danish cohort study on marijuana use (1215 participants, where 45% had smoked marijuana within the last 3 months) found 28% lower sperm concentration and 29% lower total sperm count in regular users after adjustment for confounders [93]. However, another recent cohort study (662 subfertile men) showed significantly higher sperm concentrations and total sperm counts in users [94]. Whan et al. incubated sperm with THC at therapeutic dosages (0.032 μM) and reported a 28% reduction in motility in samples with worse general parameters and no reduction in those with better parameters. Higher concentrations, similar to those for recreational use, instead caused a significant reduction in the motility of all samples (56% reduced motility in the fraction with best fertilizing potential and 28% in the poorer sperm population) [95]. In addition, THC may play a role in male libido, with augmented short-term libido and diminished ability to achieve erection [96,97]. Paternal THC use was also linked to neurobehavioral effects in offspring (decreased attentional accuracy), using an animal model [98].

Of note, an important limitation to consider is that most of these cohorts are almost entirely comprised of subfertile men smoking the psychoactive drug and not medical cannabis (CBD—cannabidiol) users. Animal studies suggest an effect on male fertility (reduced fertilization rate and number of litters) for CBD, probably due to a negative impact on sperm acrosome reaction [99].

Therapies specifically used to address MS-related fatigue, such as Modafinil, Naltrexone, 4-aminopyrine, and Amantadine (off-label), could potentially have a positive effect on male reproduction by improving sexual function. Modafinil has also been associated with a therapeutic effect on premature ejaculation in animal studies [100], some case reports [101], and a small proof-of-concept study of 55 patients [102]. There are no studies evaluating effects on fertility, but the registration studies using animal models reported no adverse effect in the prescribing information.

Concerns were raised with respect to the effect of phosphodiesterase inhibitors, used to treat erectile dysfunction, on male fertility after the report of premature acrosome reaction in vitro [103]. However, a systematic review and meta-analysis including 11 studies (1317 participants) concluded that there are no adverse effects and reported a small increase in sperm motility and morphology in infertile men [104].

In conclusion, there are several limitations to the application of the current literature to evidence-based practice. Some of them (lack of power, heterogeneity of the population, lack of standardization among different studies) are generic [105], while others (interactions with MS, disability, and DMTs) are specific to MS patients.

Table 3 summarizes the available evidence on the effect of MS symptomatic/supporting therapies on male fertility.

## 4. Concluding Remarks

An evaluation of the effect of MS on male reproductive function is complex, as it is the result of several potential factors: the effects of the disease itself on sexual function and/or fertility, the use of multiple DMTs, and symptomatic/supporting therapies.

In this review, we highlighted how limited the currently available data on each of these issues and their interactions are. On the basis of the pathogenesis of the disease and the data reported in the few observational studies available, we know that sexual function is frequently impaired and that an effect on fertility cannot be excluded. As for the medications used, few of them have a demonstrated gonadotoxic/teratogenic effect, but, for the majority of the new DMTs, data are severely lacking. As for supporting/symptomatic therapies, the available evidence, mainly extrapolated from other populations, may aid the clinician in choosing the drug with the minimum potential effect on fertility and pregnancy [111].

The still open questions indicate the research priorities in this field.

Epidemiological data on the reproductive history of affected men should be gathered on a national and supra-national level, to better understand if the potential mechanisms described in the manuscript are related to causing subfertility.

Moreover, the long-term effects of the DMTs, especially the newest immunomodulatory drugs, on sperm quality and their safety for fathering a pregnancy must be collected rigorously and reported to the scientific community. To overcome the limitation due to the relatively low number of patients affected and the great number of therapies available, multicenter studies are desirable.

Gynaecologists, andrologists, and neurologists must be up to date on the issue and able to conduct an in-depth reproductive anamnesis of males with MS. Indeed, an effective collaboration between the different specialists, similar to the networks between oncologists and fertility units [112], would greatly help us to better understand the issue and improve patient management.

## Figures and Tables

**Figure 1 jcm-10-05401-f001:**
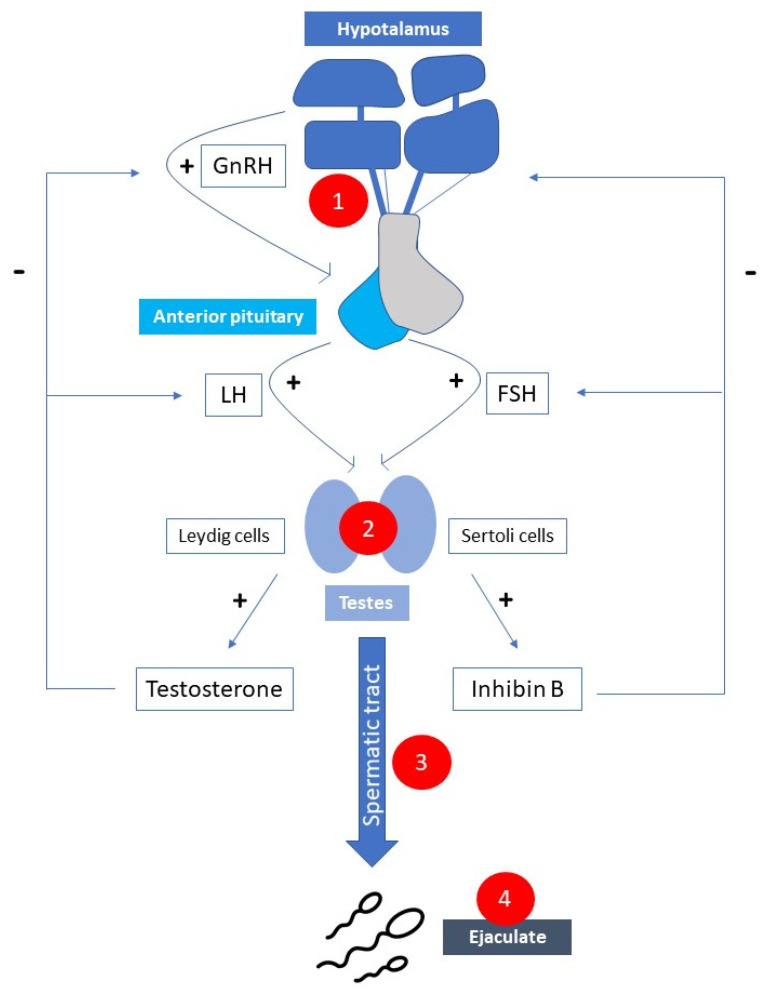
Possible causes of male subfertility correlated with multiple sclerosis. 1. Hypogonadotropic hypogonadism; 2. Hypergonadotropic hypogonadism; 3. Chronic inflammation and/or oxidative stress; 4. Erection/ejaculation dysfunction. (GnRH—gonatropin-releasing hormone; FSH—follicle stimulation hormone; LH—luteinizing hormone).

**Table 1 jcm-10-05401-t001:** Possible rationale for reduced fertility in men with multiple sclerosis.

Effect on Fertility	Pathway	Major Findings	Quality of Evidence
1. Hypogonadotropic hypogonadism	Demyelination: central neurological damage, altered hypothalamus–pituitary function	Lower mean basal serum levels for LH, FSH, and testosterone, and no significant increase with the injection of GnRH analogue (case control study: 68 men) [19] Evidence of other impaired hypothalamic functions were reported in post mortem studies in men with worse disease course (i.e., lesion of corticotropin-releasing hormone (CRH)-producing neurons, resulting in defects in adrenal function [27,28,29,30])	Case series, case control studies
2. Hypergonadotropic hypogonadism	Reduced protective role of T if low levels are present Autoimmunity	Pre-existing lower levels of T may be a risk factor: lower levels of T described in men with more progressive disease [36], transgender women at higher risk of MS than men [39]. Higher prevalence of infertility due to immunological causes was described in other autoimmune diseases [33,34].	Longitudinal association
3. Semen parameter alterations	Chronic inflammation Oxidative stress	Chronic inflammation and OS have been linked to both MS [40] and infertility [41,42].	Longitudinal association
4. Erection/ejaculation dysfunction	Demyelination: autonomic dysfunction	Suprasacral, parasympathetic, or peripheral autonomic lesion. Up to 70% of the men with MS report erection dysfunction; up to 50% alterations in ejaculation [21,22,23].	Observational studies

Abbreviations: MS = multiple sclerosis; OS = oxidative stress; T = testosterone.

**Table 2 jcm-10-05401-t002:** Summary of potential gonadotoxicity of disease-modifying therapies (DMT) for multiple sclerosis in males.

Drug	Mechanism of Action	Animal Studies	Spermatogenesis	Fathering a Child	Management in Men Trying to Conceive	FDA/EMA Warnings
Glatiramer Acetate	Shift inflammatory Th1 profile into anti-inflammatory Th2.	No effect in mice.	No mutagenic effect in in vitro and in vivo studies.	Outcomes comparable with unexposed population.	Continue use if benefits overcome risks. No washout needed.	FDA: Pregnancy category B. EMA: Decreased weight gain in infants breastfed by mothers treated during pregnancy and breastfeeding. No mention of men fathering a child.
Interferon B-1a (intramuscolar/ subcutaneous)	Anti-inflammatory (enhances the production of anti-inflammatory cytokines IL-4 and IL-10). Anti T-cell proliferation. Decreases antigen presentation.	No effects.	No alterations reported.	Outcomes comparable with unexposed population.	Continue use if benefits overcome risks. No washout needed.	FDA: Pregnancy category C. Subcutaneous is FDA and EMA approved for its safe use during pregnancy. No mention of men fathering a child.
PEGylated Interferon B-1a	Anti-inflammatory. Anti T-cell proliferation. Decreases antigen presentation. Prolonged efficacy thanks to PEGylation.	No effects.	No alterations reported.	Outcomes comparable with unexposed population.	Continue use if benefits overcome risks. No washout needed.	FDA: Pregnancy category C. No mention of men fathering a child.
Interferon B-1b	Anti-inflammatory (decreases the production of the proinflammatory cytokine IFN-gamma). Anti T-cell proliferation. Decreases antigen presentation.	No effects.	No alterations reported.	Outcomes comparable with unexposed population.	Continue use if benefits overcome risks. No washout needed.	FDA: For females, discontinue therapy if the patient becomes pregnant, plans to become pregnant, or is breastfeeding. No mention of men fathering a child.
Teriflunomide	Decreases pyrimidine synthesis. Shift Th1→Th2. Decreases antigen presentation.	In mice: reduced epididymal sperm count at high doses. Severe malformations in offspring of exposed animals.	No data, probably no adverse effect.	Limited data. One small study does not report malformations or other adverse effects in pregnancies from fathers taking Teriflunomide.	Should be avoided in men wishing to conceive.	FDA: Warning for pregnancy (may cause major birth defects). Contraindicated in pregnant women or women of childbearing potential who are not using reliable contraception. Contraception is recommended with plasma levels above 0.02 mg/L. Men trying to conceive should discontinue therapy. EMA: No recommendation of discontinuation for men trying to conceive.
Dimethyl Fumarate	Anti-inflammatory. Targets anti-oxidant mechanisms.	In mice: asthenospermia at high doses. In mice, rats, dogs: testicular toxicity at clinically relevant doses.	No data.	No adverse outcomes observed.	Not enough data for a clinical recommendation.	FDA: Pregnancy category C. EMA: No recommendation of use for pregnancy and females trying to conceive. No mention of men fathering a child.
Cladribrine	Decreases lymphocyte transit through blood–brain barrier, decreases lymphocytes subpopulations.	In mice: asthenospermia at high doses. In monkeys: reduced testicular weight at high doses.	No data. Possible damage (interferes with DNA synthesis).	No data for males.	Should be avoided in men wishing to conceive. A 6-month washout is required.	FDA and EMA warning for pregnancy: controindicated for use by pregnant women and by women and men with reproductive potential who do not plan to use effective contraception (for at least 6 months after the last dose) because of the risk of fetal harm. FDA: Discontinue therapy if a woman becames pregnant while using this drug. Controindicated during breastfeeding.
Fingolimod	Sphingosine 1-phosphate (S1P) receptor modulator. Keeps lymphocytes in secondary lymphatic organs.	No effect in mice at high doses.	No data.	No data for males. Associated with a 2-fold increased risk of a severe congenital malformation when administered in women during pregnancy.	Not enough data for a clinical recommendation.	FDA and EMA warning for pregnancy. Women of childbearing potential should use effective contraception. Washout period of 2 months required in women. No mention of men fathering a child.
Siponimod	S1P receptor modulator selective for subtypes 1 and 5. Direct action in central nervous system: inhibits demyelination and attenuates TNFα, IL-6, and IL-17 production via astrocytes and microglia.	Male rats: dose-related increase in precoital interval. Decrease in implantation sites, increase in preimplantation loss, decrease in the number of viable fetuses (at 100 times the recommended human dose).	No data.	No data for males.	Not enough data for a clinical recommendation.	FDA and EMA warning for pregnancy. Women with childbearing potential should use effective contraception during and for 10 days after stopping Siponimod. EMA: Discontinue therapy if a woman becomes pregnant while using this drug. Contraindicated during breastfeeding. No mention of men fathering a child.
Natalizumab	Anti-inflammatory. Blocks leukocyte attachment to cerebral endothelial cells.	In guinea pigs: no effects on male fertility at doses up to 7 times the clinical dose.	No effect on semen parameters and gonadal steroids after 12 months of treatment (16 men).	No data for males.	Continue use if benefits overcome risks. No washout needed.	FDA and EMA: Can be used during pregnancy when the potential benefit justifies the potential risk. No mention of men fathering a child.
Alemtuzumab	Targets CD52.	CD52s are present in male reproductive tissue. In mice: OAT, but no altered reproductive performance.	CD52 antibodies agglutinate and inactivate sperm in vitro. A study (*n* = 13) reported no effect on sperm parameters in vivo after 1, 3, and 6 months.	No data for males.	Not enough data for a clinical recommendation.	FDA: Pregnancy category C. EMA for females: Contraception while receiving the drug and for at least 4 months after the last dose is recommended. No mention of men fathering a child.
Ocrelizumab	Humanized monoclonal antibody against CD20, endovenous administration. Lyses B-cells via antibody-dependent cellular cytotoxicity (ADCC) and and complement mediated lysis.	In monkeys: no effects on male fertility at doses up to 10 times the clinical dose.	No effect on semen parameters and gonadal steroids after 12 months of treatment (16 men). Possible transient haematological anomalies in infants born to mothers exposed to other anti-CD20 B-cell-depleting antibodies during pregnancy.	No data for males.	Not enough data for a clinical recommendation.	For females: Contraception while receiving the drug and for at least 6 (FDA)–12 (EMA) months after the last dose is recommended. No mention of men fathering a child.
Ofatumumab	Humanized monoclonal antibody against CD20, subcutanenous administration. Lyses B-cells via antibody-dependent cellular cytotoxicity (ADCC) and complement mediated lysis.	In monkeys: no effect from exposures greater than 500 times that in humans at the recommended human maintenance dose of 20 mg/month.	No data.	No data for males. Possible fetal harm due to lymphopenia, possible transient haematological anomalies in infants born to mothers exposed to other anti-CD20 B-cell-depleting antibodies during pregnancy.	Not enough data for a clinical recommendation.	For females: Contraception while receiving the drug and for at least 6 months (FDA) after the last dose is recommended. No mention of men fathering a child.
Mitoxantrone	Enhances T-cell suppressor function. Inhibits B-cell function and antibody production. Remotely used in clinical practice.	No data.	Reversible azoospermia.	No data for males.	Semen cryopreservation. Washout of 6 months before pregnancy using fresh semen.	FDA: Pregnancy category D. FDA warning for pregnancy. EMA warning for fertility and pregnancy. Men in therapy had to use contraceptive measures during and for at least 6 months after the end of the therapy (4 months for women).
Cyclophosphamide (off-label)	Reduce the number of T- and B-cells. Conditioning before aHSCT.	In mice: alteration of seminal parameters with clinical treatment doses.	Azoospermia/ oligospermia, sometimes irreversible. Hypergonadotropic hypogonadism.	Increased risk of post-implantation loss.	Semen cryopreservation. Washout of 6 months before pregnancy using fresh semen.	FDA warning for fertility and pregnancy. Washout of 6 months before pregnancy using fresh semen.
Azathioprine (off-label)	Immunosuppressive activity.	In mice: temporary oligospermia and reduced number of successful mating at high doses.	No long-term effect on semen (studies in patients with IBD).	Reports of increased pregnancy complications.	Washout of 6 months before pregnancy using fresh semen.	FDA warning for fertility and pregnancy. Washout of 6 months before pregnancy using fresh semen.
Methotexate (off-label)	Immunosuppressive activity.	No data.	Reversible oligospermia.	No increase in adverse outcomes of pregnancies fathered by men taking methotrexate.	Washout of 3 months period before using fresh semen.	FDA warning for pregnancy. Washout of 3 months before pregnancy using fresh semen.
Rituximab (off-label)	Chimeric monoclonal antibody against CD20. Lyses B-cells via direct signaling of apoptosis, complement activation, and ADCC.	No effect.	No data.	No data for males. Possible transient haematological anomalies in infants born to mothers exposed to other anti-CD20 B-cell-depleting antibodies during pregnancy.	Not enough data for a clinical recommendation.	EMA: For females, contraception while receiving the drug and for at least 12 months after the last dose is recommended. No mention of men fathering a child.

Abbreviations: FDA—Food and Drug Administration; EMA—European Medicines Agency; S1P—Sphingosine 1-phosphate; IBD—inflammatory bowel disease; aHSCT—autologous hemopoietic stem cell transplantation; ADCC—antibody-dependent cellular cytotoxicity.

**Table 3 jcm-10-05401-t003:** Summary of potential gonadotoxicity of symptomatic/supporting therapies for multiple sclerosis in males.

Drug	Indication	Effect on Male Reproduction
Benzodiazepine (Aprazolam, Diazepam, Lorazepam, Clonazepam) [76,77]	Anxiety, insomnia, mood disorders. Neuropathic pain. Spasticity.	Hyperprolactinemia (may cause erection/ejaculation dysfunction and alteration of spermatogenesis). No direct effect on spermatogenesis in preclinical studies on animals.
SSRIs (Citalopram, Paroxetine, Fluoxetine, Setraline) [76,77,78,79,80,81,83] and SNRIs (Duloxetine, Venlafaxine) [84,85]	Anxiety, insomnia, mood disorders. Neuropathic pain.	All: Hyperprolactinemia (may cause erection/ejaculation dysfunction and alteration of spermatogenesis). Paroxetine: Increased sperm DNA fragmentation (human study). Fluoxetine: OAT in rats. Sertraline: OAT and increased DNA fragmentation in humans, oxidative stress in rats. Citalopram: Case reports of OAT in humans, increased oxidative stress in animal studies. Duloxetine: 1 RCT (68 healthy men) showing no effect on semen parameters, DNA fragmentation, and serum hormones after 6 weeks. Venlafaxine: Increased percentage of non-progressive motility.
NDRIs (Bupropion) [79,82,83]	Depression	Animal studies have shown no negative effect on semen parameters in rats. Lower sexual side effects in humans compared to SSRIs.
Tricyclic antidepressant (Amitriptyline) [85,86,87]	Neuropathic pain, depression, urinary symptoms.	Hyperprolactinemia (may cause erection/ejaculation dysfunction and alteration of spermatogenesis). Reduction in semen volume and motility. Oxidative stress.
CBD, THC [92,93,94,95,96,97,98,99]	Spasticity	Conflicting evidence on seminal parameters. Lowers libido. Neurobehavioral risk in offspring (animal model).
Baclofen [106]	Spasticity	Sexual dysfunction reported in men using intrathecal (not oral) Baclofen
Tizanidine	Spasticity	In rats, reduced fertility at high doses. No data for men.
Tolperisone	Spasticity	No data.
Botulinum toxin	Spasticity, urinary dysfunction.	In rats, reduced fertility at high doses. No data for men.
Phenytoin/Lamotrigine [107]	Neuropathic pain	Decreased sperm count and motility (especially Phenytoin).
Topiramate [108]	Neuropathic pain, tremor, dysesthesia.	In rats: decreased count and mobility, decreased testicular weight and testosterone levels. No data in men.
Carbamazepine, Oxcarbamazepine [88,89,90,91]	Neuropathic pain (off label)	Decrease in all sperm parameters (human study). Decreased levels of bioavailable testosterone and bioactive testosterone/LH ratio.
Gabapentin—Pregabalin [109]	Neuropathic pain	Gabapentin: Reversible suppression of testicular function in rats.
Amantadine [110]	Fatigue (off label)	Improves sexual function.
4-aminopyridine	Fatigue	No effect in animal studies. No data for men.
Modafinil [100,101,102]	Fatigue	Beneficial effect on premature ejaculation.
Naltrexone	Fatigue	No effect in animal studies. No data for men.
Phosphodiesterase inhibitors (Sildenafil, Vardenafil, Tadalafil, Avanafil) [103,104]	Erectile dysfunction	Conflicting data on semen parameters/function. Modest increase in motility and morphology in infertile men.
Anticholinergic—antimuscarinic drugs (Mirabegron, Ossibutine, Solifenacine, Tolterodine, Trospium Chloride)	Urinary symptoms	No effect on fertility in rats at high doses (registrative studies, in prescribing information).
Pramipexole	Restless legs syndrome	Not mutagenic or clastogenic in a battery of in vitro and in vivo (mouse micronucleus) assays (registrative studies, in prescribing information). No data for men.
Laxatives (Bisacodyl, Macrogol, Lactulose, Magnesium Hydroxide, Glycerin)	Constipation	Probably no effect on fertility or reproduction. No studies on reproductive toxicity in animals are available, except for lactulose (no effect on male fertility in rats in registrative studies).

Abbreviations: SSRI—selective serotonin reuptake inhibitors; SNRI—serotonin–norepinephrine reuptake inhibitors; OAT—oligoasthenoteratozoospermia; NDRI—norepinephrine–dopamine reuptake inhibitor; THC—tetrahydrocannabinol; CBD—cannabidiol.

## Data Availability

No new data were created or analyzed in this study. Data sharing is not applicable to this article.

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
