# Peer review of "Effect of Multiple Sclerosis and Its Treatments on Male Fertility: Cues for Future Research"

_jcm, 2021, doi:10.3390/jcm10225401_

Round 1

Reviewer 1 Report

The manuscript "Reproductive health issues in males with multiple sclerosis" provides a literature review of direct and associated publications on the title subject.

The review does not follow PRISMA because frankly the literature database lacks evidence for treatment of comorbidities with a background of MS. Because of the lack of evidence-based data this publication relies on associations with other medical states in the absence of MS. Thus, potential complications arising from the MS disease state on male reproductive health are missing and therefore the relevance of the associated publications is questionable.

The aforementioned then points to the authors aim which is to both provide clinicians with a useful tool for counseling. As written the extent to which the content herein will aide in counseling it is not clear.  

To fortify drug impact(s) on human male reproduction I refer the authors to the exhaustive series of publications by Drobnis EZ, Nangia AK. Adv Exp Med Biol. 2017;1034:1-4, in addition to the one referenced herein.

The authors could highlight much more the possible research priorities because clearly research data are lacking. 

Grammar and spelling errors should be checked throughout the manuscript.

Specific comments: 

It would be helpful where appropriate to include mechanism of action for a drug class. For example, Lines 140-142: Cyclophosphamide is an alkylating agent that disrupts cell division in rapidly dividing cells, e.g. spermatogenesis and cancers. This is highly signifcant.

Lines 128 - 132: Given that offspring were fathered by men with MS and taking the immunomodulatory drugs, one would not reasonably expect that reproductive health was impacted, including live birth and no adverse fetal outcomes.

Lines 160-161: Don't say 'probably has no adverse effect' if data are not pro/con. As written some may conclude the monoclonals to be safe.

Line 178: No published data exist for human male reproduction.

Line 182: define what is meant by 'supporting' and 'symptomatic'.

Line 186-187: The following statement reflects the deficiency in the current report "On note, no one of the reported human studies is focused specifically on men affected by MS." 

Line 188: Duloxitine ref. doi: 10.1111/and.14207

Lines 236-242: This paragraph is also reflective of the preceding.

In closing, the authors should greatly revise their paper to make more clear what IS and what ISN'T known.

Author Response

We would like to thank the Editor and the Reviewer for the raised comments, which we believe have significantly improved the quality of our manuscript. Kindly find below our responses to the raised comments. In the uploaded revised manuscript, the changes are highlighted in yellow. We hope that we were able to respond adequately.

Reviewer #1:

The manuscript "Reproductive health issues in males with multiple sclerosis" provides a literature review of direct and associated publications on the title subject.

The review does not follow PRISMA because frankly the literature database lacks evidence for treatment of comorbidities with a background of MS. Because of the lack of evidence-based data this publication relies on associations with other medical states in the absence of MS. Thus, potential complications arising from the MS disease state on male reproductive health are missing and therefore the relevance of the associated publications is questionable. The aforementioned then points to the authors aim which is to both provide clinicians with a useful tool for counseling. As written the extent to which the content herein will aide in counseling it is not clear. 

Reply (R) 1: we strongly agree with the Reviewer that evidence about fertility and gonadotoxicity in men with MS is severely lacking. Our main aim is to summarize the current state-of-art and underline the need for more research. We modified the abstract (see page 1, line 27) and conclusion (see page 18, line 350-367) to better clarify it, as suggested by the Reviewer.

To fortify drug impact(s) on human male reproduction I refer the authors to the exhaustive series of publications by Drobnis EZ, Nangia AK. Adv Exp Med Biol. 2017;1034:1-4, in addition to the one referenced herein.

R2: We thank the Reviewer for this suggestion, we agree that the mentioned publications explore well the challenges of obtaining accurate data on the effect of drugs on male fertility. We better clarified this concept at page 16 and 18, line and references n. 107 and 113

The authors could highlight much more the possible research priorities because clearly research data are lacking.

R3: as reported above (see R1 and R2) we modified the manuscript to make this concept more explicit, accordingly to the Reviewer’s suggestion

Grammar and spelling errors should be checked throughout the manuscript.

R4: as suggested by the Reviewer, we proofread the manuscript to improve grammar and spelling.

Specific comments:

It would be helpful where appropriate to include mechanism of action for a drug class. For example, Lines 140-142: Cyclophosphamide is an alkylating agent that disrupts cell division in rapidly dividing cells, e.g. spermatogenesis and cancers. This is highly signifcant.

R5: As suggested by the Reviewer, we added more information on pathogenetic mechanisms (see page 13)

Lines 128 - 132: Given that offspring were fathered by men with MS and taking the immunomodulatory drugs, one would not reasonably expect that reproductive health was impacted, including live birth and no adverse fetal outcomes.

R6: we agree with the Reviewer that the better proof of reproductive function is indeed fathering a child (see page 13, line 193-196).

Lines 160-161: Don't say 'probably has no adverse effect' if data are not pro/con. As written, some may conclude the monoclonals to be safe.

R7: we agree with the Reviewer and we thank him for this really relevant observation. We modified the manuscript accordingly (see page 13-14 and Table 2).

Line 178: No published data exist for human male reproduction.

R8: we modified the manuscript accordingly for better clarity (see page 13)

Line 182: define what is meant by 'supporting' and 'symptomatic'.

R9: we clarified that supporting and symptomatic therapies are all therapies that do not modify the course of the disease (as opposed to disease modifying therapies, DMTs) but have the aim of improving symptoms or complications/side effects of medications, to increase patients’ comfort and quality of life (see page 14, lines 253-256).

Line 186-187: The following statement reflects the deficiency in the current report "On note, no one of the reported human studies is focused specifically on men affected by MS."

R10: we agree with the reviewer that this is the main deficiency of the available literature and therefore of our report and it underlines the need for more focused research (see page 14, lines 257-263). However, we believe that data on the same drugs used in different diseases are a good starting point to deduce their gonadotoxicy. This, of course, does not mean that we shouldn’t focus on obtaining better data specific for MS patients, as we underlined in conclusion as well (see page 18).

Line 188: Duloxitine ref. doi: 10.1111/and.14207

R11: we added the reference suggested by the Reviewer, a newly published RCT (see page 14-15 and reference n. 86)

Lines 236-242: This paragraph is also reflective of the preceding.

R12: We agree that studies on cannabis for recreational use cannot be translated to medical cannabis use. We better explained it and we added a reference more, specific for CBD (see page 15, line 318-322 and reference n.101)

In closing, the authors should greatly revise their paper to make more clear what IS and what ISN'T known.

R13: As suggested, we modified the paper and especially “conclusions” to better explain what is known, what is not known, clarify research priorities.

Reviewer 2 Report

This study reviewed the relationship between MS and infertility. This is an interesting subject; however, the following queries should be clarified before a decision:

  1. I think the manuscript title could be changed to another comprehensive title.
  2. In the methods, it would be well if you illustrated that how many articles were found and then how many number were selected according to your inclusion or exclusion criteria?
  3. I think the introduction should be improved a bit more. You should provide more information about MS, its pathogenesis, prevalence in men and women, the age at which men and women are more sensitive, its affect to patients’ organs, etc. It is immature at current format.
  4. Is MS more frequent in older men than younger individuals? If yes, how we can explain that whether poor sperm quality is caused by MS or age?
  5. It would be good if you made tables summarizing adverse effects of MS on male reproductive function either in human or animal models. It is very important part of this manuscript.
  6. Oxidative stress and inflammation may be a main reason for poor sperm quality in MS patients, because human sperm is exceptionally susceptible ROS. You could discuss about it in more detail.
  7. I expected to see the possible mechanisms of MS effect on reproductive disabilities. It would be good if you show the possible cellular and molecular mechanisms of MS effect on male fertility by using some appropriate figures or signalling pathways.
  8. It is not a well-balanced review, because the authors have focused more on genotoxicity effect of different drugs for MS therapies on sperm quality or male fertility.

Author Response

We would like to thank the Editor and the Reviewer for the raised comments, which we believe have significantly improved the quality of our manuscript. Kindly find below our responses to the raised comments. In the uploaded revised manuscript, the changes are highlighted in yellow. We hope that we were able to respond adequately to the raised comments. 

Reviewer 2

This study reviewed the relationship between MS and infertility. This is an interesting subject; however, the following queries should be clarified before a decision:

I think the manuscript title could be changed to another comprehensive title.

R14: we agree with the Reviewer and we modified the title to better represent the content of the manuscript.

In the methods, it would be well if you illustrated that how many articles were found and then how many numbers were selected according to your inclusion or exclusion criteria?

R15: as it was also mentioned by Reviewer 1, we did not include a PRISMA figure. The available literature is severely lacking (it would have been impossible to do a meta-analysis) and the specific articles on male fertility and MS are few. The main aim of our article is to do a state-of-the-art review to underline research priorities. We modified the title accordingly.

I think the introduction should be improved a bit more. You should provide more information about MS, its pathogenesis, prevalence in men and women, the age at which men and women are more sensitive, its affect to patients’ organs, etc. It is immature at current format.

R16: as requested by the Reviewer, we implemented the general introduction to MS (see page 1-2, line 32-80).

Is MS more frequent in older men than younger individuals? If yes, how we can explain that whether poor sperm quality is caused by MS or age?

R17: While it is true that men receive diagnosis of MS later compared to women, this is multifactorial (males often avoid physicians, males with MS sometimes present a subtler disease at onset when progressive, etc) and does not mean that men with MS are elderly people; MS remains a typical disease of young adults, even though with some exceptions (paediatric forms or late/very-late onset forms). Moreover, the cohorts examined for fertility are reproductive age patients. Of course, it is more difficult to define “reproductive age” for men, since we do not have a clear ending to the reproductive window, such as menopause for women. The impact of age on male fertility is known and described but it is similarly present in the general population as well as in men with MS, so it should not give a disadvantage specific to MS patients. Moreover, the published results are usually adjusted for age.

It would be good if you made tables summarizing adverse effects of MS on male reproductive function either in human or animal models. It is very important part of this manuscript.

R18: as the Reviewer suggested we added a new Table 1 summarising how MS could affect male reproduction.

Oxidative stress and inflammation may be a main reason for poor sperm quality in MS patients, because human sperm is exceptionally susceptible ROS. You could discuss about it in more detail.

R19: we absolutely agree on the possible role of oxidative stress on male fertility in these men. We added this possible mechanism subsequent to chronic inflammation at page 4, line 152-158 (see also references n. 41, 42, 43).

I expected to see the possible mechanisms of MS effect on reproductive disabilities. It would be good if you show the possible cellular and molecular mechanisms of MS effect on male fertility by using some appropriate figures or signalling pathways.

R20 as the Reviewer suggested we implemented Figure 1, better illustrating the possible endocrine pathways causing infertility in men with MS (see Figure 1 and Table 1).

It is not a well-balanced review, because the authors have focused more on genotoxicity effect of different drugs for MS therapies on sperm quality or male fertility.

R21: we agree that our review that a more specific title was needed, as the Reviewer suggested we changed the title accordingly. Conclusions are also updated to better report what is known and what is not known on the topic.

Round 2

Reviewer 1 Report

The authors have responded appropriately to the comments associated with ver. #1 of this manuscript. The manuscript is acceptable in current form, albeit with some minor modifications for grammar.

Author Response

The authors have responded appropriately to the comments associated with ver. #1 of this manuscript. The manuscript is acceptable in current form, albeit with some minor modifications for grammar.

Reply (R) 1: We would like to thank again the Reviewer for the time and effort spent to improve the quality of our manuscript. As suggested, we proofread the manuscript for grammar and syntax.

Reviewer 2 Report

Thank you for your prompt revisions. However, Table 1 could be made more interesting. For example, you could collect all experimental and human data regarding the effect of MS on fertility potency. Afterward you could create different columns containing “models” (human or rats, rabbit, etc.), “age”, “disease duration”, major findings (e.g. testosterone deficiency, poor sperm quality, hormonal imbalance, impaired spermatogenesis, erectile dysfunction, etc.), and references. I've proposed a Table which can be seen in the attached file.

Author Response

Thank you for your prompt revisions. However, Table 1 could be made more interesting. For example, you could collect all experimental and human data regarding the effect of MS on fertility potency. Afterward you could create different columns containing “models” (human or rats, rabbit, etc.), “age”, “disease duration”, major findings (e.g. testosterone deficiency, poor sperm quality, hormonal imbalance, impaired spermatogenesis, erectile dysfunction, etc.), and references. I've proposed a Table which can be seen in the attached file.

Reply (R) 1:  We would like to thank again the Reviewer for the time and effort spent to improve the quality of our manuscript.

We expanded Table 1, accordingly to the Reviewer’s suggestion, to better explain the potential mechanisms that could reduce fertility in men with multiple sclerosis. We added a column called “major findings” to describe the currently available literature. Unfortunately, for some mechanism we do not have direct evidence, but only a plausible rationale and some suggestions coming from longitudinal studies. For example, in the text we described how lower levels of testosterone (T) were found in men with worse disease: this is not definitive proof that that MS influences T production nor that T is protective against the disease. However, the observed longitudinal associations are worth mentioning and hopefully the topic will be more specifically addressed in the future. We reported the case of transgender women, whose SM risk is similar to cisgender women (and therefore higher than it is in men) after T suppression. Other mechanisms, such as autoimmunity or oxidative stress, were implicated both in MS pathogenesis and in subfertility so their role is biologically plausible, but the link between the two has not been specifically studied in MS.

As for central demyelination with hypothalamus function alteration, this rare event was described for the hypothalamus-hypophysis-adrenal axis in post-mortem studies. There is also a case control study on 68 affected men proving lower levels of gonadotropin and lower response to GnRH in affected men versus controls (described in text and table 1). The best evidence we have are for erection/ejaculation disfunctions: they are very common in MS patients and the direct consequence of demyelization at various levels (see Table 1).

We hope that we were able to better describe the state of art on the topic and we thank again to Reviewer.

Moreover, as suggested, we proofread the manuscript for grammar and syntax.